# Role of Autophagy and AMPK in Cancer Stem Cells: Therapeutic Opportunities and Obstacles in Cancer

**DOI:** 10.3390/ijms25168647

**Published:** 2024-08-08

**Authors:** Lochana Kovale, Manish Kumar Singh, Joungmok Kim, Joohun Ha

**Affiliations:** 1Department of Biochemistry and Molecular Biology, Graduate School, College of Medicine, Kyung Hee University, Seoul 02447, Republic of Korea; kovlelochana@gmail.com (L.K.); manishbiochem@gmail.com (M.K.S.); 2Department of Oral Biochemistry and Molecular Biology, College of Dentistry, Kyung Hee University, Seoul 02447, Republic of Korea

**Keywords:** cancer stem cells, autophagy, AMPK

## Abstract

Cancer stem cells represent a resilient subset within the tumor microenvironment capable of differentiation, regeneration, and resistance to chemotherapeutic agents, often using dormancy as a shield. Their unique properties, including drug resistance and metastatic potential, pose challenges for effective targeting. These cells exploit certain metabolic processes for their maintenance and survival. One of these processes is autophagy, which generally helps in energy homeostasis but when hijacked by CSCs can help maintain their stemness. Thus, it is often referred as an Achilles heel in CSCs, as certain cancers tend to depend on autophagy for survival. Autophagy, while crucial for maintaining stemness in cancer stem cells (CSCs), can also serve as a vulnerability in certain contexts, making it a complex target for therapy. Regulators of autophagy like AMPK (5′ adenosine monophosphate-activated protein kinase) also play a crucial role in maintaining CSCs stemness by helping CSCs in metabolic reprogramming in harsh environments. The purpose of this review is to elucidate the interplay between autophagy and AMPK in CSCs, highlighting the challenges in targeting autophagy and discussing therapeutic strategies to overcome these limitations. This review focuses on previous research on autophagy and its regulators in cancer biology, particularly in CSCs, addresses the remaining unanswered questions, and potential targets for therapy are also brought to attention.

## 1. Introduction

Cancer presently stands as the primary cause of premature death and a significant obstacle to global life expectancy, with projections indicating a further increase in its prevalence over the next five decades [1]. In late-stage malignancies, the chance of complete recovery decreases, resulting in reduced survival rates. Thus, quick discovery, diagnosis, and successful treatment are critical for patients’ full recovery and improved quality of life. Patients who have undergone chemotherapy or radiotherapy frequently encounter relapses, with CSCs being identified as one of the contributing factors. CSCs were initially characterized as a subset of cancer cells capable of self-renewal and differentiation into progenitor cancer cells. These cells typically remain dormant and can re-emerge under favorable conditions, contributing to disease recurrence. Tumor heterogeneity and drug resistance are the main causes of cancer recurrence and treatment failure [2]. The formation of CSCs remains under active investigation, with various theories proposed. Some theories propose that non-cancer stem cells (non-CSCs) can dynamically convert into CSCs in response to specific stimuli, while others suggest that CSCs initially exist as a distinct population and then generate heterogeneous populations of differentiating cancer cells, with some transitioning into quiescent cell populations owing to their stem-like properties; some theories suggest these could be occurring in combination [3]. Notably, non-CSCs have been studied to adopt stem-like properties by altering their metabolic characteristics. This acquired stemness through metabolic changes is now recognized as an emerging feature of cancer, termed “metabostemness”, and plays a role in the plasticity of cancer stem cells. Metabostemness focuses on metabolic features and the biochemical pathways acquired by CSC for its maintenance and proliferation [4]. Many pathways could be exploited to maintain this stemness, with one of them being autophagy. Autophagy is involved in the reprogramming of somatic cells to induce pluripotent stem cells (iPSCs). In the early stages of induced pluripotent stem cell (iPSC) generation, stem cell pluripotency is regulated by the transcription factor Sox2. Sox2 binds to a repressive region of the mTOR (mammalian target of rapamycin) promoter, inhibiting mTOR activity. This inhibition activates autophagy, which plays a crucial role in the reprogramming of somatic cells to induce pluripotency [5]. Studies have shown that cancer stem cells might use autophagic machinery for maintaining its stemness. Given the dual role of autophagy in cancer, it remains a controversial yet promising candidate for eradicating CSCs. There is inadequate research on identifying factors that shift the balance between CSC survival and death. This review delves into the critical connections and nuanced interactions between autophagy and CSCs, even highlighting the role of AMPK in this interplay. As a major regulator of autophagy, AMPK’s influence on CSC biology can be both direct and indirect, depending on the cancer stage, type, and tumor microenvironment. This review provides an overview of AMPK’s role in this dynamic, considering the paradoxical aspects and how understanding this interplay could reveal new therapeutic targets for CSCs.

## 2. Cancer Stem Cells

Cancer stem cells are a resilient subpopulation within cancer cells and have stem cell-like properties, which are believed to stem from various factors, including microenvironmental factors, epigenetic changes, metabolic reprogramming, somatic mutations and genetic alterations, and dysregulation of stem cell signaling pathways, and yet they maintain the properties of tumorigenesis. Regardless of their proliferation potential, normal stem cells are usually in the quiescent state and remain in an inactive dormant state in the resident tissue. In contrast, CSCs are highly active and have multipotent capacity, which allows them to differentiate and migrate to different locations and form new cancer cells. Tumor cells can form a cellular hierarchy similar to that in normal tissue, in which CSCs remain at the peak and control the occurrence, malignant transformation, drug resistance, and recurrence of tumors. Compared with normal stem cells, CSCs exhibit dysregulated migration and invasion abilities, as well as an abnormal tolerance to pharmacologic and immune factors [6]. Moreover, stem cells and CSCs have specific cell markers such as PSF1 (partner of SLD five 1), CD133, and CD44, which are over-expressed in CSCs; however, normal stem cells have low expression [7], thus cells with induced expression of such markers are CSCs. In addition to their stem cell-like properties like dormancy, hypoxia, resistance to stress, EMT (epithelial–mesenchymal transition) causing migration, and metabolic flexibility, this population also seems to acquire more resilient properties like a heightened expression of drug efflux pumps, efficient DNA repair mechanisms, and seem to rely on each other. This makes them hard to kill, resulting in them being one of the biggest hurdles in clearing the cancer completely, as these properties have a threat of recurrence (Figure 1).

### 2.1. Dormancy

Dormancy occurs when a cell stops undergoing proliferation, and it can be seen as an adaptive and defensive mechanism in response to stress. CSCs can switch into a dormant or a quiescent stage in the tumor microenvironment and re-enter into a proliferative state with the emergence of favorable conditions [8]. Research findings indicate that tumor expansion initially follows an exponential track until intra-tumoral space becomes confined, triggering cells to enter a quiescent state [9]. The trigger leading to dormancy is still under study and is quite complex and multifactorial, involving several factors including epigenetic regulations, therapies, autophagy, senescence, and even hypoxia. Within the context of cancer, the classification of cancer stem cells (CSCs) further delineates distinct subpopulations, namely dormancy-competent CSCs (DCCs), cancer-repopulating cells (CRCs), dormancy-incompetent CSCs, and disseminated tumor cells (DTCs). The plasticity inherent to these cells enables them to transition between these phenotypic states in response to variations in the tumor microenvironment, facilitating an adaptive advantage in exploiting prevailing conditions [10]. 

### 2.2. Hypoxia

Hypoxia is an important factor for cancer and CSC in tumor microenvironment, which favors local invasion and drug resistance. Under hypoxia conditions, both HIF-1α (Hypoxia-inducible factor-1α) and HIF-2α expression is upregulated and promotes CSC survival and stemness. HIF-2α induces the expression of pro-survival factors such as Nanog, Oct4, and c-MYC [11]. HIF-1α activates many transcriptional factors such as snail and twist activation-associated pathways like Notch, Oct4 and Wnt Hedgehog, etc. Some reports highlighted that HIF-2α is positively linked with breast cancer and has an increased expression that facilitates the expression of tumorigenesis surface markers such as CD44 [12]. Multiple studies have demonstrated the involvement and significance of hypoxia in preserving stemness. For instance, one study revealed that hypoxia not only elevated the proportion of CD133-positive cells but also intensified the stem-like characteristics in glioblastoma cell lines [13]. Another study showed that a highly tumorigenic cell subset in a neuroblastoma cell line predominantly resides in hypoxic tumor regions and even exhibited migration towards hypoxic environments. Moreover, under hypoxic conditions, these migrating cells displayed an elevated expression of the embryonic stemness gene OCT-4 [14].

### 2.3. Drug Resistance

Drug resistance represents one of the major hurdles seen in cancer therapy, with one of the major reasons being CSCs. Drug resistance represents a significant impediment in cancer treatment, with CSCs playing a pivotal role in its manifestation. CSCs employ various strategies to counteract the efficacy of therapeutic agents, including metabolic reprogramming, upregulation of drug efflux mechanisms, augmentation of DNA repair mechanisms, and the adoption of a dormant state [15]. Studies have demonstrated that markers like CD133 are associated with drug resistance and the presence of this biomarker can upregulate the expression of ATP-binding cassette (ABC) transporters ABCG2, which is implicated in conferring resistance to first-line drugs like platinum and paclitaxel in lung cancer. Drugs like cisplatin have shown to cause DNA damage, which again leads to upregulation of ABCG2 and subsequent an increase in CD133^+^ cells [16]. In a normal stem cell population, ABC transporters are expressed as a protective mechanism, but ironically, these ABC efflux pumps also provide protection to CSCs, safeguarding them from the harmful effects of chemotherapy.

### 2.4. EMT (Epithelial–Mesenchymal Transition)

CSC induces a specific set of genes related to EMT and many adhesion factors, helping them to invade new sites [17]. Many studies have reported enhanced expression of EMT transcription factors including Snail, Twist, Zeb1-2, Slug, crypto, and vimentin in CSC [18]. Furthermore, EMT is also linked with anoikis resistance in many cancers. Anoikis is a program cell death by which detached epithelial cells from the extracellular matrix and their neighboring cells leads to apoptosis. However, cancer cell-induced EMT gene expression supports resistance to anoikis. Some studies also provided evidence that E-cadherin plays a crucial role in EMT and resist anoikis via the interaction of many proteins, such as Ankyrin-G, β-catenin, and Hippo, and the PI3K/AKT pathway [18]. Several studies have demonstrated a direct association between HIF-1, indicating that elevated levels of HIF-1α induces EMT via notch signaling, leading to increased metastatic traits in both in vitro and in vivo, thereby contributing to the intricate nature of CSCs [19,20].

### 2.5. Immunosurveillance

Another notable trait of CSCs is their effective evasion of the host’s immune surveillance. T lymphocytes are pivotal in increasing the body’s immune response against tumors but owing to the rapidly proliferating tumor harboring a pool of CSCs, natural antitumor immunity may be constrained and frequently inadequate to eradicate it completely. It was reported that CSCs lose MHC-I to silence the expression of tumor-associated antigens (TAAs). MHC class I molecules are crucial for the immune recognition of transformed cells, but a reduction in MHC class I molecules may inhibit the ability of class I MHC-restricted cytotoxic T lymphocytes (CTLs) to eliminate CSCs [21]. The tumor-microenvironment provide many cytokines and growth factors to maintain CSCs stemness in various cancers, for example, breast and oral squamous cancer [22]. Immune molecules such as IL-6, IL-8, TGF-β, NFκ, and TNF-α help in the cell proliferation, differentiation, and maturity of CSCs. Many immune cells are also associated with many signaling pathways, such as IL-6, which regulates Notch-3 and results in more metastasis in ductal breast cancer. In human glioblastoma, the deletion of IL-6 leads to cellular apoptosis and tumor growth resistance. NFκB signaling has also been reported in cancer migration and invasion, including pancreas, skin, and ovarian cancer through modulating EMT factors such as snail, twist, and slug [11].

### 2.6. Metabolic Flexibility

Cancer stem cells switch to metabolic reprogramming to maintain nutrient and energy homeostasis in a tumorigenic environment [23]. In the 1920s, a phenomenon called the Warburg effect was coined, suggesting that cancer cells consume a significantly higher amount of glucose compared to normal cells as an adaptation to meet the needs of the uncontrolled proliferation in a growing tumor [24]. Similarly, cancer stem cells have shown to increased glycolysis to maintain their stemness, notified by the overexpression of glycolysis-related genes in CSCs compared to non-CSCs [25]. It is also shown that CSCs can switch from glycolysis to OXPHOS (oxidative phosphorylation) based on niche requirements. OXPHOS also plays a crucial role in EMT by overexpressing PGC-1α in circulating cancer cells to increase mitochondrial biogenesis [26], while proteins like c-myc and Mcl1 co-operatively activate OXPHOS, which promotes drug resistance in breast CSCs [27]. In addition, since reactive oxygen species (ROS) are a byproduct of OXPHOS, and appropriate amounts of ROS can help in CSC maintenance and differentiation, OXPHOS provides an ideal framework for metabolic reprogramming [28]. However, metabolic heterogeneity may occur within the same tumor, where both glycolysis and OXPHOS can be active. If glycolysis is inhibited, CSCs may switch to OXPHOS to evade cell death, illustrating the complexity of CSCs’ metabolic adaptations based on their microenvironment and differentiation stage [29]. Now, to mitigate the ROS production as a result of increased OXPHOS, counteractive mechanisms like employing potent antioxidant systems are taken [30]. ROS stimulation subsequently triggers nuclear factor-erythroid 2-related factor 2 (NRF2), which activates the antioxidant defense system and maintains redox homeostasis by reprogramming the metabolism in CSCs, ultimately maintaining the stemness of CSCs [31]. In many reports, higher levels of unsaturated fatty acid are found in cancer stem cells, which inhibit the stearoyl-COA desaturase 1 (SCD1) and acetaldehyde dehydrogenase 1 A1 (ALDH1A1) that maintain stemness and growth of many tumors by inhibiting apoptosis [32]. Several such studies have been conducted to distinguish CSCs’ specific metabolic mechanisms, which can serve as potential biomarkers and targets for cancer therapy.

To sum up, all these properties of CSCs are fundamentally interconnected and is the reason why CSCs are hard to kill, making them one of the major reasons of relapse. Taking into account these variables, multiple targets could be employed to effectively eliminate cancer stem cells (Figure 1).

### 2.7. Regulatory Mechanisms in CSCs vs. Non-CSCs vs. Normal Cells

As discussed above, CSCs tamper with normal cellular mechanisms to ensure survival. Examples of these alterations include the management of oxidative stress, DNA repair processes, enhanced drug resistance, autophagy, and other critical regulatory pathways. For example, in normal cells, ROS levels are regulated to support necessary signal transduction for cell survival [33], while elevated levels of ROS are seen as a hallmark of cancer cells and CSCs as a result of hyper metabolism [34]. Consequently, CSCs have developed a strategy to upregulate antioxidant systems by manipulating specific transcription factors, such as NRF2, which in turn regulates the expression of genes involved in ROS detoxification. Increased NRF2 expression has been observed in cervical CSCs [35], while increased expression of target genes of NRF2 was seen in breast CSCs [36], colon CSCs [37] and glioblastoma CSCs [38]. Similarly, the DNA damage response (DDR) is essential for maintaining genomic integrity and preventing cells from succumbing to mutagenic events. Research has shown that cancer cells experience elevated levels of DNA damage due to replication stress, leading to a domino effect of hyperproliferation. Elevated DNA damage calls for enhanced DDR, which is observed in various CSCs. For instance, a study showed that lung CSCs exhibit heightened DNA repairing capabilities to cisplatin-induced DNA damage compared to lung cancer cells [39]. Prostate CSCs (PCSCs) showed increased γH2AX levels and induced G2/M arrest in PCSC, causing resistance to etopside, suggesting increasing DDR in these CSCs [40]. The aforementioned characteristics are not only typical of CSCs but also reflect their ability to manipulate various pathways, contributing to their immortality. This review specifically addresses autophagy, one of the major regulatory processes that has been extensively studied, offering valuable insights into the potential vulnerabilities of CSCs.

## 3. Autophagy

### 3.1. Autophagy in Normal Cells vs. Cancer Cells

Autophagy is a recycling process that plays a crucial role in maintaining homeostasis and is often upregulated is stress conditions like starvation, hypoxia, DNA damage, ER (endoplasmic reticulum) stress, or pathogen infection [41,42,43] to prevent the cell from undergoing unrepairable damage. There are many factors that regulate autophagy in cells, among which mTOR and PI3K pathways are the main pathways that contribute to autophagy regulation. Under nutrient starvation, AMPK is known as a negative regulator of mTOR and activates autophagy. Unc-51-like kinase (ULK) is the only core protein with serine/threonine kinase activity, which triggers the formation of an initiation complex. ULK1, FAK family kinase-interacting proteins (FIP200), autophagy-related proteins like ATG13, and ATG101 form a complex, translocate to the membrane, and form autophagosomes [44]. The ULK1 complex phosphorylates, the class III phosphatidylinositol-3-kinase (PI3K), vacuolar protein sorting-34 (Vps34), and beclin-1 (Vps34-beclin1), initiate the formation of phagophore [45]. Vps34 is a catalytic subunit involved in the formation of two complexes: Complex-I for autophagosome nucleation and Complex-II for endosomal trafficking. Vps34 forms Complex-I with Vps15-Beclin-1-ATG14L, and Complex-II with UVRAG-beclin1-Vps15. Later, in the presence of autophagy-related proteins (ATG), the lipidation of LC3B with ATG9 and ATG4 occurs, and ATG7, ATG3, and ATG5-ATG12-ATG16L1 transform LC3B to LC3B-II [46]. The formation of autophagosomes involves several factors including the SNARE (soluble NSF attached protein receptor) complex, RABS (Rab-GTPase), HOPS (homotypic fusion and vacuole protein sorting) complex; and later recruitment of LC3B-II, resulting in fusion with lysosomes. Finally, auto-phagolysosome degrade the sequestered materials inside the autolysosomes via hydrolase [47].

### 3.2. Tumor-Suppressive Effects of Autophagy in Early Cancers

Autophagy functions by identifying targets like impaired cellular organelles, protein clusters, and pathogens within cells and play a crucial role in meeting metabolic demands during conditions such as nutrient scarcity, genotoxic stress, lack of growth factors, and low oxygen levels [48]. Autophagy is also involved in immunosurveillance, where the immune system continuously eliminates potentially cancerous cells before they develop into tumors [49]. Furthermore, autophagy in stem cells contributes to preserving their distinct characteristics, such as differentiation and self-renewal capabilities [50]. However, during excessive genotoxic oxidative stress, along with some other extrinsic and intrinsic factors, autophagy is dysregulated. Defects in autophagy can cause a buildup of ubiquitylated protein aggregates such as p62/SQSTM1 and dysfunctional organelles. This results in several functional consequences, including the accumulation of reactive oxygen species (ROS) which promote cellular damage, reduce stress tolerance, and make cells more susceptible to tumor initiation and pathogen invasion [51,52]. Moreover, the absence of the autophagy-associated gene Beclin-1 (BECN1), which plays a crucial role in phagophore formation, is noted to show decreased autophagy and increased cell proliferation in various types of human breast, prostate, and ovarian cancers [53,54,55]. The knockout of other proteins associated with autophagy like ATG5, ATG7, and UVRAG have shown a higher incidence of cancerous growth [56,57]. The increased occurrence of cancer observed with dysfunctional autophagy strongly supports the idea that autophagy acts in a tumor-suppressive manner against the development of malignancies in breast, prostate, pancreatic, lung, liver, and ovarian cancers [58]. Several investigations also indicate that autophagy can be regulated by certain tumor suppressive pathways. Notably, p53 is known to inhibit autophagy at basal levels, but when triggered by cellular stress like DNA damage, it activates numerous genes that promote autophagy, including DRAM1 (damage-regulated autophagy modulator 1) [59], Tsc2, and AMPK [60,61]. 

### 3.3. Tumor Protective Properties of Autophagy in Established Cancers

These tumor-suppressive functions, when hijacked by cancer cells, happen to empower the pre-malignant cells to escape genotoxic stress and inflammation, in turn promoting tumorigenesis [62]. Advanced tumors often exhibit heightened tolerance to harsh microenvironments, facilitating cellular adaptation and the acquisition and maintenance of stem cell-like properties. Several studies have suggested that autophagy plays an opposite role in advanced established tumors [63]. Additionally, enhanced autophagy is often seen to be upregulated in cancer stem cells, which has opened new research areas of interest. The summarized version is graphically described in Figure 2.

#### 3.3.1. Relationship between Tumorigenicity and Autophagy in CSCs 

High heterogeneity of cancers, consisting of different populations of cancer cells, CSCs, and cancer-associated neighboring cells, such as cancer-associated fibroblasts (CAFs), causes complexity in cancer biology and difficulties in identifying a single molecular target for anti-cancer methods. An increasing number of studies have shown that autophagic homeostasis between these heterogenous cancers and associated cell populations is an intrinsic feature of the phenotype of the cancers and results in resistance to chemotherapy [3,64,65,66,67]. For instance, in a human breast CSC study, two different populations of cancer cells, breast CSCs and breast non-CSCs, were isolated and showed different autophagic activity and correlated characteristics. The breast CSCs (CD24^−^/CD44^+^/ALDH^+^) is characterized by enhanced autophagic activity, increased resistance to chemotherapy, and mesenchymal phenotype, but breast non-CSCs have the opposite characteristics: decreased autophagy, higher sensitivity to chemotherapeutic reagents, and an epithelial, rather than mesenchymal, phenotype. Importantly, autophagy inhibition has been shown to negatively affect the migration (metastasis) and stemness of CSCs and the secretion of a key tumor-promoting and CSC-maintaining cytokine, IL-6. In fact, autophagy is upregulated in the mammospheres, and both BECN1 and ATG4 are needed for their maintenance and expansion. Similarly, autophagy has been demonstrated to play an important role in the maintenance of a variety of CSCs in pancreatic cancer [68], liver cancer [69], osteosarcoma [70], ovarian cancer [71], and glioblastoma [72]. Interestingly, in the case of leukemia, autophagy appears to function differently between AML (Acute Myeloid Leukemia) and CML (Chronic Myeloid Leukemia) as well as the cancer stage [73,74]. Earlier studies have demonstrated the pathophysiological significance of autophagy in the survival, tumorigenic potential, and chemoresistance of AML [75]. By contrast, the loss of autophagy in hematopoietic stem cells has shown an expansion of a AML progenitor population, especially the invasive myeloproliferative cells in bone marrow [76]. Moreover, autophagy promotes leukemic cell death by degrading master oncoproteins, such as promyelocytic leukemia-retinoic acid receptor α (PML-RAR) and breakpoint cluster region-Abelson kinase (BCR-ABL) [77,78]. In CML, autophagy genes such as ATG4, ATG5, and BECN1 are reported to be upregulated and impairing this autophagy, either by silencing ATG7 or ATG4 or by treating with HCQ, sensitizes CML to chemotherapeutic treatments [79,80] (Figure 3).

#### 3.3.2. Autophagy in Hypoxic Environments

Numerous studies suggest that in advanced cancers, autophagy supports tumor survival and proliferation under severe stressors like hypoxia and nutrient scarcity [81], particularly in the central regions of solid tumors, where hypoxic conditions prevail and tumor hypoxia is evidently associated with excessive angiogenesis, metastasis and resistance to chemotherapy [82,83,84]. In fact, studies indicate that HIF stimulates the expression of stem cell markers like OCT4, SOX2, NANOG, MYC, and miR-302 in cancer cells, implying that hypoxic niches may have a significant role in transforming non-cancer stem cells into cancer stem cells [85]. Hypoxia-induced autophagy, particularly driven by HIF-1α, not only supports the survival of liver CD133^+^ CSCs, but also maintains the balance between non-stem pancreatic cancer cells and pancreatic CSCs, highlighting its dual role in both cell survival under hypoxic conditions and the conversion of non-stem cancer cells into stem-like cells [86]. 

#### 3.3.3. Influence of Autophagy in EMT

Another pattern seen in advanced tumors is epithelial–mesenchymal transition (EMT). EMT involves cellular changes leading to the loss of adhesion and polarity, occurring in embryonic development, wound healing, tissue regeneration, organ fibrosis, and tumor progression [87]. The interplay between EMT signaling and CSC phenotypes is evident, with CSCs notably possessing migratory abilities, supporting their metastatic potential [88,89], and it is seen that autophagy is being linked to this interplay. Recent discoveries provide insights into the role of autophagy in Glioblastoma and uncover a novel role for the autophagy regulators DRAM1 and p62 in regulating migration and invasion in cancer stem cells [90]. In colorectal cancer, EMT is regulated by the SOX2-β-catenin/Beclin1/autophagy signaling axis in colorectal CSCs [91].

#### 3.3.4. Signaling Pathways Involved in Autophagy Dependent CSCs

Much effort has been made to understand the molecular mechanisms of autophagy-dependent CSCs’ maintenance and function. A mouse model study has shown that EGFR/Stat3 and TGF-β/Smad signaling is required for autophagy functions in two distinct breast cancer stem-like cells CSCs (ALDH+ and CD29^high^/CD61^+^, respectively) [92]. Upon the knockout of RB1CC1 (encoding FIP200/ATG17, a one of key component of ULK1 complex), the inactivation of the EGFR-STAT3 signaling axis was observed, and consequently, there was an impairment of ALDH+ breast cancer stem cells’ (BCSCs) tumorigenicity. On the other hand, autophagy inhibition leads to a decrease in TGF-β expression, concomitantly inactivating Smad signaling, which is indispensable for the CD29^high^/CD61^+^ breast CSCs phenotype. Moreover, the experiments on a triple-negative breast cancer model showed that the inhibition of autophagy decreases IL-6 secretion via the JAK2/STAT3 pathway [93], which is closely related to the number of CD44^+^/CD24^−^ breast CSCs. It supports the idea that the IL-6-JAK2-STAT3 signal transduction pathway could play an important role in the conversion of non-CSCs into CSCs. The FOXO transcription factor family is another candidate contributing to the autophagy–CSC network [45]. FOXOs are well known to participate in a fundamental transcription program for the homeostasis of stem cell in both embryos and adults [94], although the underlying molecular mechanism still needs to be defined. The knockdown of FOXO3 results in the increased CSC self-renewal capacity in prostate [95], glioblastoma [96], ovarian [97], breast [98], liver, and colorectal cancer [99], whereas leukemia needs FOXO3 for stem cell maintenance [100]. In the autophagy context, FOXOs have been reported to mediate the transcription of autophagy and its related genes, including ATG5, ATG8, ATG12,ATG14, BECN 1, ULK1, LC3, GABARAPL1, and BNIP3; interestingly, cytosolic FOXOs appear to take part in the regulatory mechanism of autophagy [101]. 

#### 3.3.5. Autophagy in Drug Resistance

Despite the progress in radiation and chemotherapy targeting fast-growing cells, it is evident that CSCs, residing in a dormant state and developing resistance to standard treatments, play a crucial role in tumor recurrence. An enhanced DNA damage response is a key factor in cancer progression and resistance to chemotherapy. For instance, CD133^+^ tumor cells in gliomas show a heightened activation of DNA damage checkpoints and repair DNA damage more efficiently after radiation compared to CD133^−^ cells and may contribute to tumor recurrence post-radiation [102,103]. Tumor dormancy also leads to multidrug resistance (MDR) via the increased expression of ABC transporter proteins, avoidance of apoptotic pathways, and increased expression of CSC markers linked to drug resistance [15,104]. Recent studies have posed the involvement of autophagy in drug resistance [15]; for example, CD133, an important CSC marker, was discovered to enhance cisplatin resistance in gastric cancer stem cells (GCSCs) by boosting cell proliferation, anti-apoptotic activity, and autophagy capability through the activation of the PI3K/AKT/mTOR signaling pathway, and CD133 knockdown was able to reverse cisplatin resistance [105]. These studies collectively indicate the impact of autophagy on drug resistance and how autophagy inhibitors can enhance the sensitivity of cancer stem cells to specific drugs.

Another emerging area of drug resistance research focuses on the crosstalk between cancer cells and mesenchymal stem cells (MSCs), a subset of stem cells that play a dual role in both inhibiting and promoting cancer growth. The secretome of these cells includes cytokines, growth factors, microvesicles (MV), and extracellular matrix (ECM) proteins, which maintain the tumor microenvironment (TME) and support cancer progression [106]. A study conducted in 2020 demonstrated that MVs secreted by MSCs could manipulate non-small cell lung cancer’s (NSCLC) TME and thus activate autophagy, supporting cancer survival [107]. A similar study revealed that bone marrow MSCs contribute to the progression of multiple myeloma, exacerbating cell migration as a result of activated autophagy, and drug resistance to doxorubicin and Velcade through the release of microvesicles (MVs) [108]. Although this area of research is still emerging, it holds significant potential as a target in cancer biology.

Tumor-suppressing elements are inhibited by mTOR, leading to the initiation of autophagy and the inhibition of cancer formation, while, in contrast, oncogenes can be stimulated by mTOR, class I PI3K, and AKT, leading to the inhibition of autophagy and the promotion of cancer development [48]. However, this process can be highly exploited in the cancer environment. So, autophagy in cancer is highly context dependent and can play complex roles depending on the type, stage, and the environment. In the early stages of cancer development, autophagy can suppress tumorigenesis by removing damaged cellular components and preventing the accumulation of genomic instability, whereas in established tumors, cancer cells can hijack the autophagic machinery to support their survival and growth. Table 1 provides a comprehensive summary of the role of autophagy across various types of cancer (see Table 1).

### 3.4. Mitophagy in CSCs

#### 3.4.1. Mitophagy and Tumorigenesis

Mitochondria are organelles determining either cell proliferation/survival or cell death, mainly through cellular energy production, reactive oxygen species (ROS) production, and apoptotic activation [111]. Cellular energy production in mitochondria inevitably generates ROS, which can damage DNA, membrane structures, and cellular proteins, to induce cell senescence and death. Therefore, the mitochondria homeostasis program to remove non-functional (long-lived or damaged) mitochondria is essential for all cellular functions, which is composed of two arms, mitochondria fission/fusion and mitochondria-specific autophagy and mitophagy. Mitophagy blocks dysfunctional mitochondrial function and limits ROS production via lysosome-dependent mitochondrial degradation. Moreover, it can eliminate surplus mitochondria as an adaptive response to stresses, such as hypoxia and nutrient deprivation, to balance cellular energy requirement with oxygen consumption and concomitant ROS production. Mitophagy requires specific adaptors to recognize and load the mitochondria into the autophagosome. Accumulating reports have demonstrated these mitochondria-specific cargo receptors and their underlying mechanisms [112]. The most well-known mitophagy-initiating machinery includes PTEN-induced putative kinase-1 (PINK1) and E3 ubiquitin ligases Parkin. Mitochondrial membrane depolarization in dysfunctional or overloaded mitochondria activates PINK1 on the outer mitochondrial membrane (OMM), where it phosphorylates and recruits Parkin. Activated PINK1/Parkin phosphorylates and ubiquitinates a number of OMM proteins, such as Mitofusin-2 (Mfn2) and the voltage-dependent anion channel-1(VDAC-1). Then, various mitochondria cargo receptors, including Optineurin (OPTN), NDP52, and p62/Sqstm1 recognize and load these ubiquitinated protein-tagged mitochondria on autophagosomes by interacting with LC3 via their LC3-interacting region (LIR) [113,114,115,116]. In addition to PINK1/Parkin, pro-apoptotic BH3-only proteins, Bcl2-interating protein 3 (BNIP3), and BNIP3L (Nix) play an important role in mitophagy in response to hypoxia [117]. Similar to other mitochondria cargo receptors, BNIP3 and BNIP3L have an LIR motif near the amino terminus. In hypoxia, the expression of BNIP3/BNIP3L is induced at both the transcriptional and post-translational levels [117,118,119]. Phosphorylation of the serine residue in the LIR motif in BNIP3/BNIP3L promote LC3 binding followed by mitophagy. Notably, BNIP3L is shown to be ubiquitinated by Parkin and to promote PINK1/Parkin-induced mitophagy [120,121], indicating the molecular connection between the PINK/Parkin axis and BINP3/BINP3L [122]. Also, it is worth noting that both BNIP3 and BNIP3L interact with Rheb [123,124], a small GTPase required to promote lysosomal localization and activation of mTORC1, a key growth-promoting as well as autophagy-inhibiting kinase complex [125]. BNIP3 binding repressed Rheb activity, disrupted the BNIP3-Rheb interaction and promoted mTORC1 activity and cell growth, which is consistent with the growth-suppressive role of BNIP3 via its inhibitory interaction with Rheb [123]. It suggests that BNIP3/BNIP3L holds cell growth and simultaneously activates mitophagy to eliminate damaged mitochondria in the context of low oxygen conditions as a cell protection program. Like BNIP3/BNIP3L, FUN14 domain-containing 1 (FUNDC1) also promotes hypoxia-induced mitophagy. FUNDC1 functions as a mitochondria cargo receptor through its LIR motif and its activity is regulated in a manner dependent on phosphorylation [126]. Autophagy-initiating protein kinase ULK1 phosphorylates the serine residue close to the LIR motif on FUNDC1 to increase FUNDC1-LC3 interaction, leading to mitophagy [127]. FUNDC1 is also phosphorylated by oncogenic Src tyrosine kinase and casein kinase 2 (CK2), but these phosphorylations are demonstrated to inhibit LC3 interaction [127,128]. In contrast, mitochondrial serine/threonine phosphatase phosphoglycerate mutase 5 (PGAM5) dephosphorylates CK2-dependent phosphorylation on FUNDC1 to promote LC3 interaction and mitophagy [128]. Interestingly, PGAM5 is also documented to function in mitophagy by promoting PINK1 accumulation [129]. Collectively, these reports suggest that various mitophagy-inducing machinery, such as PINK1/Parkin, BNIP3/BNIP3L, and FUNDC1, are closely interconnected and integrate a variety of mitochondrial stress in order to fine-tune cellular mitophagy activity. 

In the context of tumorigenicity, mutations and deletions of key mitophagy molecular components PINK1 (*PARK6*) and Parkin (*PARK2*) genes are frequently observed in various cancer types, including bladder, breast, lung, ovarian, colon, and glioblastoma [130,131,132,133]. Consistently, a number of studies on various PINK1/Parkin-null mouse models have demonstrated that PINK1/Parkin is closely involved in the susceptibility to spontaneous hepatocellular carcinoma (HCC) [134], the sensitivity to irradiation-induced lymphomagenesis [135], and the increase in tumor burden and metastasis in oncogenic K-Ras-driven pancreatic cancer [136,137]. Notably, PINK1/Parkin also functions in angiogenesis through the regulation of HIF-1α stabilization [130,137,138], in which Parkin ubiquitinates HIF-1α to promote its degradation. Indeed, low Parkin level was reported to be correlated with high HIF-1α level and poor prognosis in breast cancer. Mechanism studies have demonstrated that mitochondrial damage blocked cell cycle progression in a PINK1/Parkin-dependent manner, while the loss of Parkin or PINK1 increased the rates of cell division [139]. Similarly, deletions in BNIP3 genes and epigenetic silencing of BNIP3 promoters are also extensively documented in a number of cancers, such as breast, gastric, pancreatic, liver, lung, and hematological malignancies [140,141,142,143,144] and are shown to be closely associated with chemoresistance and poor prognosis [141,145]. The oncogenic Src kinase-dependent regulatory mechanism for FUNDC1-LC3 interaction [127] also indicates that FUNDC1-dependent mitophagy plays a role in Src-driven tumor cell migration and invasion [146]. However, it should be noted that there is conflicting evidence demonstrating the role of FUNDC1 in cancers. Bioinformatic analysis showed that the decrease in FUNDC1 was associated with increased ROS signaling and metastasis, particularly in lung cancer, but another study reported elevated FUNDC1 with increased tumor cell proliferation, migration, invasion, and the resulting poor prognosis for breast cancer patients [146,147].

#### 3.4.2. The Role of Mitophagy in CSCs 

Although metabolic reprogramming is known to be a key feature of CSCs to orchestrate their self-renewal capacity, stemness, tumorigenicity, and the resistance to chemotherapeutic reagents [3,148], the role of mitochondria-dependent metabolism is complicated and not simply defined in CSCs. Initially, it has been believed that many cancer cells rely on glycolysis with a dramatic decrease in oxidative phosphorylation (OXPHOS) in mitochondria (Warburg effect) [149,150]. Consistent with this notion, it has been shown that glioma CSCs are driven by a glycolytic reprogramming, exhibiting more fragmented mitochondria than neuronal stem cells and downregulation of mitochondrial respiratory activity in the CSCs [151]. However, growing evidence has indicated the opposite results in many different cancer types [152]. 

Mitochondria have highly dynamic structures to maintain homeostasis by undergoing fission (regulated by MFN1,2 and OPA1) and fusion (by DRP1) [153]. Notably, mitochondrial fission and fusion play an important role in regulating mitophagy [154]. Indeed, DRP1-driven mitochondrial fragmentation is reported to function in the acquisition and maintenance of CSCs [155]. Brain tumor-initiating cells (BTICs), a type of neuronal CSCs, activates mitochondrial fission through CDK5-dependent DRP1 activation to prevent cell death and, thus, to maintain self-renewal and growth [156]. DRP1 activation in BTICs correlates with poor prognosis of glioblastoma. Similarly, mitophagy plays an important role to deal with ROS by removing the damaged mitochondria, which is required for the maintenance of CSCs and their tumorigenicity [157]. Importantly, mitochondrial fission and DRP1 phosphorylation by ERK2 are required for oncogenic RAS transformation [158,159]. DRP1-dependent fission also plays crucial roles in oncogenic B-RAF transformation in melanoma [160,161]. Consistently, inhibition of DRP1 phosphorylation blocked tumor cell growth in xenografts, while pancreas-specific deletion of *Drp1* inhibited progression of pancreatic ductal adenocarcinoma in model mice [158,162]. As expected, the combination of Trametinib (for ERK1/2 inhibition) with hydroxychloroquine (HCQ), for autophagy inhibition) was shown to very effectively reduce tumor cell growth and tumor volume in the pancreatic cancer patients [163,164]. Interestingly, loss of PINK1-dependent mitophagy is also sufficient to dramatically decrease the efficiency of iPSC reprogramming from mouse embryonic fibroblasts [165], indicating that mitophagy is directly responsible for determining the fate of stem cells. Consistently, in hematopoietic stem cells, mitophagy is shown to maintain the regenerative capacity of old hematopoietic stem cells [166].

### 3.5. Importance of Context Specificity in Cancer

Investigating the context-specific effects of autophagy in different cancer types, stages, and microenvironments can provide insights into the mechanisms underlying their dual roles and guide the development of more effective therapeutic interventions. For instance, RAS mutations are detected in various cancers and rank highest among pancreatic cancers, followed by lung and colon cancers [167]. Additionally, studies have demonstrated the potential link between Ras mutations and the emergence of stemness traits in cancer cells [168]. Elevated basal autophagy levels are observed in cancers induced by RAS mutations, in fact suggesting autophagy dependency [169]. This “autophagy addiction” has been recognized as a vulnerability in RAS mutant tumors, given the absence of approved drugs to directly target RAS mutations pharmacologically [170]. Research has demonstrated that inhibiting autophagy decreases cell proliferation in pancreatic ductal adenocarcinoma (PDAC) and extends the survival of mice [171,172]. Additionally, the inhibition of ERK has been observed to promote the activation of AMPK and beclin-1, while reducing mTORC1 signaling and consequently impacting the transcription of genes associated with autophagy in pancreatic cancer [163]. Another study discovered that blocking MEK1/2 promotes the LKB1→AMPK→ULK1 signaling pathway, resulting in autophagy, and combining MEK1/2 inhibition with autophagy blocking demonstrates potential antitumor benefits against PDAC. Similar findings were observed in melanoma with (Neuroblastoma-RAS) NRAS mutations and colorectal cancer with BRAF mutations, providing additional validation for this study [164]. Moreover, in a study investigating the role of autophagy in PDAC, markers for hypoxia, CSCs, and autophagy were found to be co-expressed in patient-derived PDAC tissues suggesting that that enhanced autophagy levels enable CSC survival under hypoxic starvation conditions and inhibition of autophagy disrupted this intricate balance of autophagy in CSCs, shifting survival signaling towards cell death [173]. It is also shown that the involvement of autophagy in PDAC tumor development is fundamentally linked to the status of p53 [174]. Rosenfeldt et al. demonstrated that, in mice lacking key autophagy genes such as ATG5 or ATG7, early-stage pancreatic lesions arise but fail to advance to cancer. Conversely, in mice harboring mutated Kras and deficient in p53, the absence of autophagy does not hinder tumor progression; rather, it accelerates tumor formation. These findings suggest that the efficacy of autophagy and autophagy inhibition may vary depending on the genetic context and stage of cancer [174]. Conversely, the absence of p53 in Ras-induced lung cancer cells promotes increased tumor progression compared to those with intact p53. However, this phenomenon leads exclusively to the development of benign tumors termed oncocytomas, which exhibit an excess of dysfunctional mitochondria [175]. However, autophagy dependence seems to vary among different cell lines possibly due to different and number of mutations in Ras, thus sensitivity to autophagy would change accordingly and these instances underscore the contextual nature of autophagy’s involvement in cancer, depending upon the specific oncogenic drivers leading to cellular transformation. 

## 4. Regulators of Autophagy in Cancer

### 4.1. AMPK and Its Role in Autophagy

Autophagy regulators not only govern the process of autophagy but also hold considerable significance as targets within the realm of cancer biology. mTOR is one of the major regulators of autophagy and maintains balance between cell growth and autophagy. In any stress conditions, including nutrient deprivation, mTOR is suppressed and thus the balance is shifted towards autophagic cascade [176]. AMPK, being the upstream regulator of mTOR, is also involved in this regulation. AMPK is able to phosphorylate various sites in ULK-1, which is an important component for the initiation of formation of autophagosomes. AMPK is a cellular energy sensor that is activated when the energy levels are low, or to be specific, when AMP/ATP ratio is increased; thus, it targets specific certain physiological processes that trigger the production while correspondingly decreasing energy-consuming processes. AMPK also regulates whole body energy metabolism in addition to intracellular homeostasis, and a dysfunctional trigger or an imbalance readily leads to diseases like type 2 diabetes and cancer [177]. Thus, AMPK has become one of the representative molecules linking cell metabolism and cancer; therefore, AMPK has become an important target for treating metabolic diseases as such. 

### 4.2. Involvement of AMPK in Regulating Autophagy within the Context of Cancer

AMPK signaling regulates autophagy via various molecular interactions [178]. Cancer cells under nutritional deficiency, hypoxia, and oxidative stress activates AMPK signaling, which leads to mTOR inhibition, thereby relieving the suppression of autophagy [179,180]. AMPK activates autophagy via direct phosphorylation of ULK1 upon the upstream signal by LKB1. AMPK activates ULK1 by phosphorylation at Ser555, Thr574, Ser467, and Ser637, which initiates the autophagosome formation. AMPK negatively regulates mTOR by the TSC1 and TSC2 complex that results in the phosphorylation of Raptor at Ser722 and Ser792 and regulation of the autophagy activation [181,182]. Under normal conditions, mTOR suppresses autophagy, particularly mTORC1, which regulates the signals related to nutrient availability and energy status of the cell. Therefore, AMPK regulates autophagy via mTOR complex-1 or by the phosphorylation of ULK1 in CSCs depending on the tumor microenvironment and cancer type. ULK1 independently activates the Beclin1-VPS34-Atg14L complex by the phosphorylation of beclin1 at Ser15 and ATG14 at Ser29. Furthermore, ULK1 recruits and activates VPS34 in phosphatidylinositol-3 (PI3)-kinase complex-I on the phagophore membrane and forms the autophagosome [183].

Likewise, mTOR has emerged as a significant factor in cancer biology due to its connection with AMPK and its involvement in cell survival, proliferation, and angiogenesis through the PI3K/Akt/mTOR axis. mTOR inhibition by rapamycin showed to enrich the CD133^+^ population in liver cancer [184]. Similar results were seen in colorectal cancer (CRC), where MSC (mesenchymal stem cells) secretions led to activated AMPK/mTOR signaling, causing an increase in migration and proliferation of CRC [185]. The PI3K/Akt/mTOR pathway is deregulated in various tumors, positioning mTOR as a potential therapeutic target, as confirmed by multiple studies. Inhibiting mTOR has been shown to suppress the growth of CD133^+^ pancreatic cancer stem cells and their sphere-forming ability, thereby confirming its role in self-renewal and CSC maintenance [186]. mTOR, along with Notch signaling, was found to play an important role in cellular proliferation in gastric CSCs and the inhibition of either of those signalling pathways would restrict gastric adenocarcinomas [187].

Under stress conditions such as hypoxia and nutritional deficiency, Hif-1 regulates AMPK activation, which results in the induction of autophagy via BINP3/Beclin-1 or by mTOR inhibition. The autophagy regulator LETM1 (Leucine-zipper and EF-hand-containing Transmembrane protein 1), mitochondrial inner membrane protein, maintain mitochondrial morphology and the mitochondrial Ca+/H+ ion channel [188]. LETM1 controls the growth of CSCs in colorectal cancer (CRC) through ROS/AMPK/mTOR signaling [189]. AMPK is also reported to phosphorylate several core components of mitochondrial autophagy/mitophagy, including PAQR3 on T32 [190], ATG9 on S761 [191], Beclin1 on S91 and S94 [192], VPS34-associated protein RACK1 on T50, and VPS34 on T133 and S135 [193], under nutritional deficiency. In Hematopoietic stem cells (HSCs), LKB1-AMPK axis plays an important role in maintaining homeostasis and quiescence [194]. Several proteins associated with epithelial–mesenchymal transition (EMT) and stemness are overexpressed in breast cancer stem cells. Although AMP-activated protein kinase (AMPK) may not directly regulate these properties, it plays a role in modulating other regulators of stemness and EMT. For instance, MYC activation drives the transition from quiescent cells to proliferating cells by inhibiting YAP/TAZ. This strains cellular energy, activating AMPK, suggesting that AMPK indirectly helps the transition of dormant breast cancer cell to amplifying cells [195]. Moreover, AMPK knockout MEFS was not able to induce autophagy despite glucose deprivation, suggesting the functional importance of AMPK in autophagy [196] (Figure 3).

The impact of AMP-activated protein kinase (AMPK) and autophagy on cancer stem cells (CSCs) is multifaceted and context-dependent, influenced by factors such as the tumor microenvironment, disease stage, and the intricacies of signaling cascades. The dynamic interplay between AMPK, autophagy, and CSCs underscores the complexity of tumor biology [195] and therapeutic responses, making them both potential targets for cancer therapy but also potential drivers of tumor aggressiveness. Further investigation into the molecular mechanisms governing these interactions is essential for the development of targeted therapies and a nuanced understanding for therapeutic intervention.

## 5. Therapeutic Approaches

In light of these viewpoints, researchers have devised diverse approaches to address this challenge, yielding promising outcomes that are undergoing clinical evaluation, while some are still undergoing scrutiny. Here are some recent strategies alongside their respective constraints.

### 5.1. Metformin

A potential link between the usage of metformin and a lowered risk of cancer among diabetes patients was observed in a study conducted in 2005. This study revealed a 23% reduction in cancer risk associated with metformin use [197]. Individuals with reduced metabolism, obesity, hyperglycemia, and hyperinsulinemia are more likely to develop cancer [198]. This can be easily connected to the fact that metabolic alterations seen in malignant cells include increased glucose absorption and glycolysis. In various animal models with altered metabolic profiles, caloric restriction has been shown to enhance longevity and reduce cancer incidence. Metformin, an anti-diabetic medicine, has been demonstrated to mimic the effects of calorie restriction by influencing cellular metabolism via a variety of pathways compared to patients on other treatments. This regulation reduces energy-demanding activities within cells, including cellular proliferation [199]. 

The anti-cancer properties of metformin can follow more than one pathway, which is likely to co-interact. One of the mode of actions of metformin is that it inhibits the oxidative phosphorylation, specifically the mitochondrial complex I, and causes ROS production, causing the cancer cells dependent on oxidative phosphorylation for survival to be killed [200], which causes ATP imbalance and leads to AMPK activation. Activated AMPK by metformin not only inhibits protein synthesis, but also inhibits the Warburg effect, including aerobic glycolysis, thus causing glucose deprivation for cancer cells, inhibiting tumor growth by activating AMPK, inhibiting mTOR, and suppressing HIF-1 alpha (which is partially activated by mTOR), which promotes glycolysis-related gene expression [201,202]. Although the data are limited, a few studies have shown that metformin inhibits the folate-dependent one-carbon metabolic pathways, including the de novo maintenance of intracellular nucleotide pools essential for DNA repair and synthesis in various breast cancer cell lines [203] and pancreatic cell lines [204], contingent upon AMPK activity. Metformin-induced activation of AMPK can suppress Wnt/β-catenin by signaling via AMPK, thus diminishing the unregulated cell proliferation in colon carcinoma cells [205] and breast cancer [205]. Metformin is shown to be effective even via pathways independent of AMPK. Metformin was able to abolish mTORC1 Rag in a GTPase-dependent manner rather than AMPK/TSC1/2 [206]. 

As mentioned above, epithelial–mesenchymal transition (EMT) and mesenchymal–epithelial transition (MET) are well acknowledged as a critical process underpinning cancer stem cell production, metastasis, and recurrence. Given its significance, focusing on understanding and targeting this transition holds promise for better predictive results and early intervention techniques in cancer eradication [207]. Metformin has been shown to impede EMT via hedgehog (Hh), Wnt, and transforming growth factor beta (TGFβ) pathways in many cancers including breast, cervical and NSCLC cancers [208,209,210]. NF-κB plays a pivotal role in cancer stemness by fostering a pro-inflammatory milieu, hindering apoptosis, and promoting cell proliferation in cancer stem cells [211]. According to research, metformin may selectively target cancer stem cells (CSCs) over non-stem cancer cells (NSCCs), as it inhibits inflammatory pathways such as NF-κB and IL6 in CSCs, resulting in selective CSC death and tumor growth reduction. Metformin disrupts the inflammatory feedback loop in CSCs by inhibiting NF-κB nuclear translocation and STAT3 phosphorylation [212]. Metformin is also studied to serve as a negative regulator of the mevalonate (MVA) pathway, reducing CSCs in CRC by blocking protein prenylation activities. Unlike cholesterol synthesis inhibition, protein prenylation inhibition greatly lowers CSCs, suggesting a possible therapeutic target. Furthermore, metformin’s effects on CSCs and the MVA pathway are linked to AMPK activation and mTOR inhibition, indicating a complex mechanism of action [213] (Figure 3).

From a technical standpoint, since autophagy could support the growth of cancer stem cells, the activation of AMPK by metformin might trigger this process, potentially aiding cancer cell growth. However, the bulk of evidence suggests otherwise. Metformin’s overall impact on cancer stem cells tends to be suppressive due to its varied effects on cellular metabolism and signaling pathways. It is essential to note that most observational studies have primarily examined cancer incidence concerning metformin’s influence on the development of cancer, which is distinct from its effects on the proliferation of established tumors [214]. Therefore, despite the initial appearance of contradiction between metformin’s induction of autophagy and its anti-cancer properties, the cumulative effect of metformin on cancer stem cells is inhibitory, ultimately contributing to decreased tumor growth and progression.

### 5.2. Autophagy Inhibitors

Given that cancer stem cells depend on autophagy as survival mechanism in harsh conditions, the use of autophagy inhibitors as a weapon to target this “dependence” has peaked interests in cancer therapy. In addition, the observation that some cancer cells exhibit heightened autophagy following anti-cancer drug therapies has further highlighted the importance of incorporating autophagy inhibitors into therapeutic strategies. In many cancers, a combination of autophagy inhibitors with cancer drugs showed more beneficial results. In fact, Chloroquine (CQ) and Hydroxychloroquine (HCQ), known autophagy inhibitors, are extensively used in clinical trials that target CSCs [215]. In gastrointestinal stromal tumors, a combination of imatinib and chloroquine (CQ) promotes tumor cell apoptosis [216]. Autophagy inhibition via the knockdown of ATG12 and ATG7 showed reduced CSC stemness and cell death. ATG5 knockdown reduces the expression of stem cell-associated markers Sox2, Oct4, and tumorigenic potential in CD44^+^ ovarian CSCs [217]. In another example, hepatic Axn2^+^CD90^+^ CSCs and the knockdown of ATG3 and ATG7 reduce hepatocyte growth factor (HGF) expression and CSCs’ stemness [218]. Another research showed that bevacizumab, a standard chemo drug in glioblastoma treatment could increase the chemotherapy resistance of glioblastoma cells by elevating their autophagy levels, achieved through the suppression of the Akt-mTOR signaling pathway. Moreover, the use of autophagy inhibitors amplified the toxicity of the drugs [219]. Chloroquine was also seen to decrease the ability of doxycycline resistant breast cancer cells and sensitize them to doxycycline [220]. Similar studies have shown that Salinomycin’s inhibition of the autophagic flux in breast cancer stem cells disrupts their stemness [221]. In chloroquine (CQ)-mediated inhibition of autophagy in sorted ALDH^hi^ cells from the breast cancer cell line MDA-MB-231, a decrease in stem cell properties and an enhanced sensitivity to chemotherapy drugs (doxorubicin, DOX, and docetaxel, DTXL) were noted. Moreover, a nanoparticle-based drug delivery system was applied, which notably extended the half-life of circulation and increased drug accumulation within tumor tissues and cancer stem cells (CSCs), resulting in significant tumor suppression in breast CSCs [222]. Another study demonstrated that carboplatin alone induced significant DNA damage repair in TNBC cells with intact BRCA1. However, the addition of CQ effectively attenuated this response, particularly in cancer stem cells (CSCs) and thus suggested that combining CQ with carboplatin may offer greater efficacy compared to carboplatin alone in TNBC patients, potentially mitigating issues such as relapse, metastasis, and chemoresistance [223]. In glioblastoma stem cells, MST4 kinase was upregulated by irradiation, which leads to the increased phosphorylation of ATG4B, more autophagy flux in GSCs, support of stemness, and increased tumorigenicity in vivo [224]. The inhibition of ATG4B phosphorylation and autophagy inhibition promotes the therapeutic effects of radiation in a GBM. Another interesting study has been conducted trying to use the autophagy dependence of PDAC, where they found that ERK pathway inhibition in KRAS mutated PDAC and increased autophagy dependence even more, making it easy to target it with autophagy inhibitors.

Hydroxychloroquine (HCQ), a chloroquine analog with one-third the toxicity, differs from chloroquine by the addition of a hydroxyl group and also inhibits autophagy [225]. Similar to CQ, HCQ has also been explored in cancer therapeutics as an individual or combination therapy, especially in patients with established tumors [226]. In fact, several studies are in clinical trials with pre-justified preclinical studies. One study demonstrated that HCQ could enhance the anti-cancer efficacy of an HDAC inhibitor vironostat in patients with advanced solid tumors [227]. Similar results were seen in another study, which is under phase II trial and showed a synergistic effect with HCQ and temozolomide in patients with advanced tumors and melanoma [228]. Another study reported a phase I trial evaluating the efficacy of hydroxychloroquine and bortezomib in patients with relapsed or refractory multiple myeloma compared to bortezomib alone [229]. Patients with metastatic pancreatic adenocarcinoma have known to have heightened autophagy and autophagy inhibition using HCQ as a monotherapy was reported by Wolpin et al. [230]. Another report demonstrated that Bcr-Abl inhibition by c-Abl specific tyrosine kinase inhibitors (TKIs) induces autophagy, and it hypothesized that autophagy protects CML stem cells, contributing to disease persistence in CML patients. The use of HCQ increased TKI induced cell death, indicating that targeting autophagy could be a therapeutic strategy and is under clinical trials [79]. All these studies suggest that the combination therapy using autophagy inhibitors with anti-cancer drugs sensitized cancer cells to chemotherapy more effectively.

### 5.3. In Silico/Computational Models

Computational models have been developed to explore the interactions and the impact of AMPK activation or inhibition in various tumor microenvironments. Recent study conducted has shown that mTORC2 inhibition holds potential in cancer treatment but may have adverse effects on insulin sensitivity; however, combining mTORC2 inhibitors with AMPK activators could enhance their anti-cancer effects [180,231]. Pineda et al. conducted another study focusing on MAGE-A3/6, an E3 ubiquitin ligase specific to cancer cells. They identified it as a pivotal oncogene and proposed its potential utility as a biomarker for identifying patients who could derive significant benefits from AMPK agonists. This is because MAGE-A3/6 targets AMPK for ubiquitination and degradation, leading to the suppression of autophagy [232].

### 5.4. Peptide/Small Molecules

Due to the limited number of autophagy inhibitors approved by the FDA, researchers have explored alternative approaches and perspectives to target autophagy. One of the strategies involves the development of peptides designed to function as competitive ligands in critical protein interactions involved in the autophagic process. Pavlinov et al. developed a peptide using high throughput cellular Nanobret assay that can specifically target interaction between Beclin 1−ATG14L, which is essential for the formation, appropriate localization, and functioning of the VPS34 Complex, thus essentially inhibiting autophagy [233]. Gray and colleagues employed a similar strategy using a Scanning Unnatural Protease Resistant (SUPR) mRNA display to create SUPR4B1W, a cyclic peptide with cell permeability that acts as an inhibitor of the LC3 protein, disrupting LC3 protein–protein interactions and thereby inhibiting autophagosome maturation. Their findings showed that this disruption sensitized cells to cisplatin and resulted in almost total tumor growth inhibition when SUPR4B1W was coupled with carboplatin in vivo [234]. LL-37, a human antimicrobial peptide, has been shown to inhibit autophagy in pancreatic cancer via altering the AMPK-ERK-mTOR signaling pathway. Furthermore, it has exhibited a reduction in myeloid-derived suppressor cells and an increase in T cells, demonstrating its potential to combat immunosuppression and ability to reprogram the tumor microenvironment, which is frequently linked with cancer stem cells [235]. Another study has focused on a biomarker for hypoxia induced transmembrane enzyme, namely carbonic anhydrase (CA) IX in cancer cells and have developed peptide based self-assembled nanofibers of (CA) IX, which undergoes size augmentation in response to decrease in ph. The endocytosis of (CA)IX together with these peptide nanofiber leads to intracellular acid vesicle damage and blocks protective autophagy. Furthermore, combining these nanofibers with doxorubicin exhibits synergistic effect demonstrating antitumor, anti-angiogenesis, and antimetastatic effects [236]. Despite potential degradation by proteases, ongoing research seeks to mitigate this limitation. Nonetheless, peptides and small molecules remain promising therapeutic modalities in cancer treatment due to their tendency for fewer off-target effects and almost no side effects.

### 5.5. Natural Compounds

In natural product space, there is a continued search for safer alternatives aimed at targeting cancer stem cells (CSCs). A study conducted by Mai et al. showed that ginsenoside F2 derived from ginseng elicited apoptosis and using autophagy inhibitors further increased cell death in breast cancer stem cells [237]. Another study has demonstrated that resveratrol and curcumin can induce autophagy and cause autophagic cell death in breast cancer stem cells and colorectal cancer cells [238,239]. Treatment of glioblastoma cells with resveratrol showed similar outcomes, eliciting autophagy-induced apoptosis. Parallel administration of an autophagy inhibitor alongside resveratrol led to reduced sphere formation and diminished levels of CD133^+^ and OCT4^+^ cells, markers indicative of glioblastoma cancer [240]. Resveratrol has also been demonstrated to inhibit growth of colorectal cancer cells by enhancing ROS, which in turn induced autophagy via the upregulation of (LC3-II) [241]. Resveratrol (RSV) exerts a strong and rapid influence on metabolism by inhibiting mTOR and S6K while activating AMPK. Additionally, RSV activates the JNK pathway, leading to the upregulation of genes involved in both the early and late stages of autophagy, and it has also shown to induce autophagy in both imatinib-sensitive and imatinib-resistant chronic myelogenous leukemia cells [242]. Rottlerin, a plant-derived chemotherapeutic agent, demonstrates encouraging efficacy in stimulating autophagy through AMPK activation, subsequently leading to apoptosis in prostate cancer stem cells [243]. Given that Rottlerin is also a selective inhibitor of protein kinase C δ (PKC-δ), it suppresses NF-κB signaling, resulting in the activation of autophagy in breast, pancreatic, and colon cancer cells [244]. IGF-BP2 plays a role in tumor progression, metastasis, and the proliferation of cancer stem cells. It has also been identified as a biomarker associated with resistance to chemotherapy in non-small cell lung cancer (NSCLC). Trichostatin A (TSA) has demonstrated the ability to counteract this resistance to cisplatin by promoting autophagy [245].

## 6. Conclusions

Cancer stem cells are becoming a hotspot for scientists working in cancer therapy as this population plays an important role in relapse and drug resistance. The study of cancer stem cells (CSCs) is an evolving field with ample opportunities for new discoveries, as highlighted in the previous section. Identifying vulnerabilities within CSCs has become a priority for researchers seeking to eradicate this population. Autophagy, a cellular recycling mechanism utilized by CSCs for survival and stemness maintenance, represents one such vulnerability. Numerous regulators govern this process, offering potential targets for cancer therapy including AMPK, which stands as an upstream of autophagy. Current studies, however, have certain limitations and gaps. This review has shed light over such loopholes; for instance, research indicates a clear relationship between AMPK–autophagy, and autophagy–CSC maintenance. Theoretically, it is expected for AMPK activation to promote CSC growth, but metformin, an AMPK activator, inhibits CSC growth, suggesting the involvement of unknown signaling pathways. Another unresolved question involves determining the factors that dictate the dual role of the autophagy–AMPK axis in cancer and how this understanding can be leveraged to disrupt the balance between CSC survival and apoptosis. Drug resistance, which poses a significant threat to CSC proliferation, may involve mechanisms beyond increased efflux pump activity or enhanced DNA damage response due to high drug concentrations. Insufficient research has been conducted on the secretome of CSCs, which is known to contain various cytokines that facilitate metastasis and the conversion of non-CSCs to CSCs. This area holds considerable importance in the study of CSC biology. Researchers have devised strategies to sensitize CSCs to various drugs, optimizing efficacy by considering specific contextual factors by using computational models to discover CSC specific properties. However, the complexity of these phenomena presents challenges in cancer diagnosis. This review comprehensively examines obstacles and therapeutic avenues in targeting CSCs by elucidating the interplay between autophagy and its regulators.

## Figures and Tables

**Figure 1 ijms-25-08647-f001:**
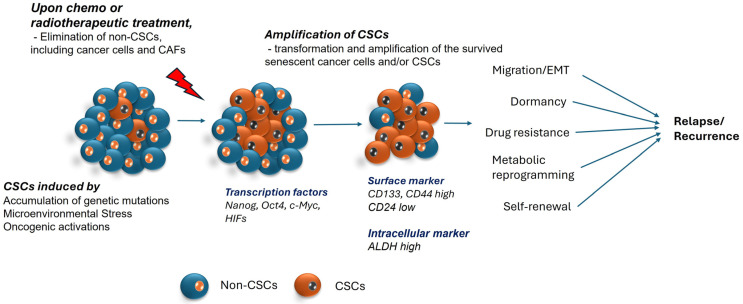
Illustration of cancer stem formation and factors related to recurrence. Cancer stem cells are known to originate as a result of somatic mutations, microenvironmental triggers, and epigenetic modifications. This population has the ability to survive after chemotherapy as it develops drug resistance, altered metabolic signals, and the capacity to enter quiescent states to avoid cell death. Transcription factors like Nanog, Oct-4, c-Myc and surface markers like CD133, CD44, and many more are demonstrated to be biomarkers of CSCs. These characteristics allow them to withstand extreme environments and lead to relapse.

**Figure 2 ijms-25-08647-f002:**
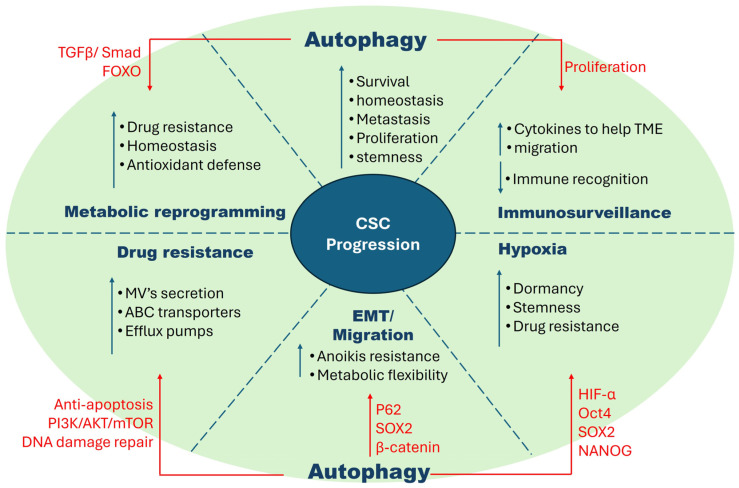
Characteristics of cancer stem cells and factors that influence their proliferation. CSCs possess robust characteristics that make them difficult to eradicate. These include hypoxia tolerance, EMT/migration, drug resistance, metabolic flexibility, and immune surveillance. These traits interact and share pathways, increasing their complexity. Autophagy, a highly studied area in CSCs, has been shown to exacerbate these characteristics by enhancing specific metabolic pathways as shown in the figure.

**Figure 3 ijms-25-08647-f003:**
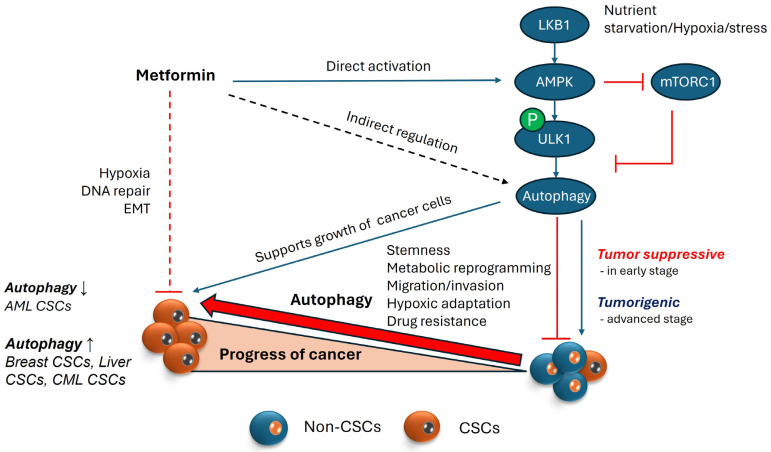
Autophagy and its role in maintenance of CSCs. AMPK is a cellular energy sensor activated during nutrient scarcity and stress, and it regulates autophagy by phosphorylating ULK1, a protein involved in autophagy initiation. Autophagy, a cellular recycling process, maintains homeostasis under stress conditions but exhibits a dual role in cancer. In the early stages of cancer, autophagy acts as a tumor suppressor by inducing cell death in potential cancer cells. However, under severe stress, cancer cells exploit autophagy to avoid starvation. In the advanced stages of cancer, where the cancer stem cell (CSC) population is growing, autophagy supports CSC properties by supplying nutrients in hypoxic environments and upregulating proteins involved in stemness/functionality of CSCs, which aid migration and invasion for cancer metastasis. Metformin, an AMPK activator, is known to inhibit cancer stem cells (CSCs) by targeting proteins involved in hypoxia, DNA repair, and epithelial–mesenchymal transition (EMT). This inhibition can occur with or without AMPK involvement. Theoretically, AMPK activation via metformin could support cancer cell survival by promoting autophagy. However, because autophagy’s role in cancer is context-dependent, the direct relationship between metformin’s inhibition of CSCs and AMPK-induced autophagy remains unclear. (Upward pointing arrows represent upregulation or increase; Downward pointing arrows represent downregulation or decrease).

**Table 1 ijms-25-08647-t001:** Table stating different cancers and role of autophagy in its maintenance and proliferation.

Cancer	Role of Autophagy	Ref.
Pancreatic cancer	Autophagy maintains survival in hypoxia, a crucial factor in PDAC Maintaining stemness by upregulating NOTCH, WNT, and SHH pathways	[68]
Liver cancer	Essential for LCSC maintenance. Autophagy helps tolerate CSCs under hypoxia and nutrient starvation conditions	[69]
Osteosarcoma	Helps in tumorigenesis, drug resistance, and tumor recurrence	[70]
Ovarian cancer	Essential for chemoresistance and self-renewal via FOXA2	[71]
Glioblastoma	Causes drug resistance The autophagy-associated factors DRAM1 and p62 regulate cell migration and invasion in glioblastoma stem cells	[72,90]
Breast cancer	Autophagy regulator Beclin 1 helps mammospheres formation and maintenance of breast CSCs	[64]
Bladder carcinoma	Maintains spheres forming ability in side population (SP) Causes drug resistance	[109]
Colorectal cancer	Protects colorectal CSCs from apoptosis causing chemoresistance SOX2-β-catenin/Beclin1/autophagy signaling axis regulates stemness and EMT in CRC	[91,110]

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
