# Peer review of "Role of Autophagy and AMPK in Cancer Stem Cells: Therapeutic Opportunities and Obstacles in Cancer"

_ijms, 2024, doi:10.3390/ijms25168647_

Round 1

Reviewer 1 Report

Comments and Suggestions for Authors

In this manuscript, the authors summarized and discussed the role of autophagy and its regulation in cancer stem cells. Generally, the manuscript is well-written. Some concerns as shown below.

1. This manuscript focuses on the role of autophagy in cancer stem cells. However, it is important to address the different regulatory mechanisms in cancer stem cells, non-cancer stem cells, and normal cells.

2. It is necessary to draw a graph indicating the regulatory mechanisms of autophagy in cancer stem cells.

3. It is informative to include a table to summarize the role of autophagy in different cancers. Related references should be included in the table.

4. Mitophagy is discussed in the current manuscript. However, it is not well discussed, and less information is provided. Again, whether the regulatory mechanisms in cancer stem cells and non-cancer stem cells are not clear.

5. AMPK and mTORC1 are two key regulators of autophagy. It is better to summarize and discuss mTORC1 signaling in cancer stem cell autophagy regulation.

6. The English of this manuscript should be further revised for easy reading and understanding. Some sentences are too long, but not clearly indicate the meanings. In addition, abbreviations are not correctly used. Full names and abbreviations appear several times at same time. Full names of some abbreviations are missing. Such irregular writings should be checked and revised throughout the manuscript.

Comments on the Quality of English Language

The English of this manuscript should be further revised for easy reading and understanding. Some sentences are too long, but not clearly indicate the meanings. In addition, abbreviations are not correctly used. Full names and abbreviations appear several times at same time. Full names of some abbreviations are missing. Such irregular writings should be checked and revised throughout the manuscript.

Reviewer 2 Report

Comments and Suggestions for Authors

This manuscript addresses a critical aspect of cancer biology by exploring the role of autophagy and AMPK in CSCs and proposing potential therapeutic strategies. It is well-conceived and provides significant insights, but there are areas that require improvement for better clarity and impact.

1) The abstract should provide a succinct overview of the review's purpose, methods (how were the articles selected?), key findings, and conclusions. Revise the abstract to clearly state the main objective, the role of autophagy and AMPK in CSCs, and the therapeutic implications.

2) The introduction should start with the global impact of cancer, leading to the challenge of CSCs, and then introduce autophagy and AMPK as critical factors in CSC biology. Restructure the introduction for better clarity and flow.

3) The manuscript would benefit from the inclusion of a diagram showing the potential or suggestive relationship between autophagy, AMPK, and CSC maintenance, and how these pathways can be targeted therapeutically (maybe improving Figure 2??). What are the distinct roles of autophagy and AMPK in the regulation of cancer stem cell differentiation and resistance to chemotherapy, and how can these pathways be selectively targeted to enhance therapeutic efficacy?

4) The discussion should not only summarize the findings but also provide insights into future directions, potential clinical applications, and unresolved questions in the field. Enhance the discussion and conclusion to emphasize the significance of targeting autophagy and AMPK in CSCs for cancer therapy.

Reviewer 3 Report

Comments and Suggestions for Authors

In the review paper entitled " Role of autophagy and AMPK in cancer stem cells: therapeutic opportunities and obstacles in cancer" authors explained details about the cancer stem stem cells and how the autophagy and AMPK signaling modulating cancer stem cells. I would suggest to accept the review for publication with minor changes.

1. I would request to add one schematic representing various factors regulating cancer stem cells properties.

2. Cancer secreted extracellular vesicles regulates its drug resistance,  immune surveillance, autophagy etc. It would be appreciated if you can include how the CSC secreted extracellular vesicles modulates autophagy.

Round 2

Reviewer 1 Report

Comments and Suggestions for Authors

The authors have addressed all the major concerns, and the manuscript has been improved.